# "Everything Will Be Fine": A Study on the Relationship between Employees' Perception of Sustainable HRM Practices and Positive Organizational Behavior during COVID19

**Amelia Manuti [1], Maria Luisa Giancaspro [1,*], Monica Molino [2], Emanuela Ingusci [3], Vincenzo Russo [4], Fulvio Signore [3], Margherita Zito [4] and Claudio Giovanni Cortese [2]**

1   Department of Education, Psychology, Communication, University of Bari, Palazzo Chiaia Napolitano Via Crisanzio 42, 70121 Bari, Italy; amelia.manuti@uniba.it
2   Psychology Department, University of Turin, Via Verdi 10, 10124 Turin, Italy; monica.molino@unito.it (M.M.); claudio.cortese@unito.it (C.G.C.)
3   History, Society and Human Studies Department, University of Salento, Via di Valesio 24, 73100 Lecce, Italy; emanuela.ingusci@unisalento.it (E.I.); fulvio.signore@unisalento.it (F.S.)
4   Department of Business, Law, Economics and Consumer Behaviour "Carlo A. Ricciardi", IULM University, Via Carlo Bo 1, 20143 Milan, Italy; vincenzo.russo@iulm.it (V.R.); margherita.zito@iulm.it (M.Z.)
*   Correspondence: maria.giancaspro@uniba.it

**Abstract:** Sustainable human resource management practices represent one of the main organizational strategy to survive and to prosper within the fast-moving current scenario. According to this view, sustainability is strictly linked to the consideration of the unique and distinctive value that each human resource means for organizations. The recent COVID19 pandemic is having a serious impact on organizations and on their employees, it is profoundly changing the working modalities, mainly introducing smart working practices that were showed to have significant consequences on workers' wellbeing. This study aims to investigate employees' perception of sustainable HRM in the frame of the COVID19 emergency, exploring if and to what extent perceptions of involvement and organizational support together with individual coping strategies associated with organizational change could influence positive organizational behaviors, namely organizational engagement and extra-role behavior. The research involved 549 participants who completed a self-report online questionnaire encompassing psycho-social measures of the abovementioned variables. Results confirmed the important role played by sustainable HRM practices both for the capitalization of human resources and of organizational performance in a time of great uncertainty and global crisis. Implications for theory and HRM practice development were also discussed.

**Keywords:** sustainable HRM; coping with change; extra role behavior; smart working; organizational citizenship behavior

## 1. Introduction

The radical changes experienced by individuals, communities and organizations on occasion of the COVID19 pandemic outbreak have completely redesigned the world as we knew it. A specific focus on organizations, both in private and public contexts, shows the huge economic, cultural, and managerial difficulties these are still facing to survive on the market and to go on providing goods and services that in most cases are necessary to customers.

Forced remote working, staff reduction, temporary production interruption, in some cases closing of the plants because of the imposed lockdown are only some of the experiences that have literally

shaken work processes, roles distribution, job demands and performances. Employees and managers were totally unprepared to manage this cognitive, affective and behavioral burden that asked both to display/develop soft skills, e.g., resilience, optimism, innovation, and adaptability, precious to cope with the unexpected. In this perspective, the COVID19 pandemic could be read as a crisis situation for many organizations that consequently have been forced to make rapid decisions and to perform proper behaviors in order to manage catastrophic happenings, in terms of social and financial implications, reassuring employees and customers about the safe and effective style adopted to secure sustainable competitive advantage [1]. Accordingly, a crisis, like the one as the pandemic, could not be simply considered from a mere financial point of view. It entails very important implications in terms of human resource management, people being the first and most important capital of the organization, especially during these very challenging moments. In this perspective, a crisis features itself as a disruptive event that could negatively influence employees' perceptions and attitudes towards the organization, ruining its reputation for both internal and external stakeholders. Yet, human resources managers have a strategic role when a crisis strikes. They represent the bridge between employer and employees, they serve goals and interests, that are neither organizational nor personal but something between the organization's management and employees. Therefore, through dedicated practices and policies they are called to convey messages to employees that could reinforce the person/organization relationship, engage and motivate the workforce toward a successful performance and positive behavior.

Moving from these reflections, the present study aims to describe the role of this important managerial function of any organization, focusing not simply on the actions and interventions that are formally planned and carried out with reference to people management (recruitment, training and development, compensation, information sharing, etc.,) rather on the employees' related perceptions and evaluations that have been proved to be significantly related to organizational resilience in these difficult moments of crisis like the one currently experienced.

In view of the above, the study intended to give an empirical contribution to this discussion, illustrating results coming from a survey conducted with a group of Italian workers experiencing remote working during the lockdown of phase 1 of the pandemic in March 2020. The main aim is to explore relationships between employees' HRM practices perceptions, individual coping with change strategies, and positive behaviors such as extra-behaviors and organizational engagement. Therefore, the paper opens with a literature review of the main scientific contributions to the positive relationship between HRM practices' perceptions and organizational success and performance. Further, the paper describes the study conducted as for participants, aims, variables, measures, and main results. Finally, a critical discussion of the implications for theory development and professional practices is presented.

## 2. Conceptual Framework and Research Hypotheses Development

### 2.1. Sustainable HRM and Positive Organizational Behavior

In the past decades, the term sustainable HRM has received growing attention within the scientific HR-related literature, mostly focusing on the key topic of organizational social responsibility and the relationship with stakeholders, inside and outside the organizational borders [2–9]. The label "sustainable HRM" represents the attempt to apply a people-oriented perspective to the development of HRM policies and practices in order to enhance organizational performance both in terms of efficacy and efficiency.

Accordingly, extant research on sustainable development showed that the paradigm of sustainability could be used beyond a wider ecological and socio-economic domain and applied to the investigation of the psycho–social features of the working context that best contribute to create conditions to promote individual as well as organizational development [10,11]. This view is attuned with a wider perspective within the work and organizational psychology underlining the crucial role played by people in organizations, namely by the intangible human capital that could concretely

contribute to secure organizational performance [12]. In this perspective, human resource management policies and practices have become a priority to support, develop, enhance people at work [13].

This evidence has been largely supported also by research coming from the positive organizational behavior (POB) perspective, studying the subjective and contextual conditions that allow individuals and communities to feel good at work, to perform better, and finally to generate competitive advantage for organizations [14]. Accordingly, a consistent number of scientific contributions showed that supportive HRM systems might increase employee motivation and engagement, significantly impacting on the productivity and performance [15–20].

Parallel to these evidences, a steadily growing body of research started exploring employees' perceptions of HRM, in order to better design "sustainable" people management systems that could be attuned with workers' expectations and could consequently foster positive behaviors. Several theoretical speculations were developed.

Drawing on the social exchange theory [21–23] and focusing on the employee perspective, prior research investigated individuals' perceptions about the fulfilment of the mutual expectations implied in the employment relationship, maintaining that, where positive, these perceptions might determine levels of satisfaction, motivation, work engagement, leader-member exchange, perceived organizational support, and organizational citizenship behaviors [24–30]

On the other hand, espousing a managerial perspective, organization studies mostly concentrated on the impact that effective practices could have on organizational performance, therefore on its efficiency and economic competitiveness [31,32]. This interest is evident in the plurality of labels featuring this research perspective, ranging from high-performance work systems to innovative HRM to high-commitment HR. A common trait to all these studies is the assumption that HR practices mainly serve to improve performance. To this purpose, HRM strategically uses human potential for organizational success, aiming to influence employees' abilities (A), motivation (M), and opportunity to contribute (O)—the so-called AMO model—maximizing results. According to this view, very little consideration is given to the concrete implications that a positive employment relationship between person and organization could have also for organizations [33,34].

However, more recently, supported by abundant scientific evidences highlighting the (also economic) benefits of a people-based approach to HRM [35], even scholars belonging to this body of research started maintaining the need to overcome a control approach in favor of a commitment approach to people management, pointing out a set of HR practices that could best impact on individual and organizational positive behaviors, keeping interdependence between mutual benefits and investments [36,37]. In fact, it is evident that employees interpret HRM in different ways [38] and their perceptions of HRM are likely to be more strongly related to their attitudes and behaviors than the actual HR practices [39].

In view of the above, within the past decades, scholars and practitioners revised most approaches to people management focusing on the enhancement of what has been called "high-commitment" policies and practices, including employee involvement in managerial decisions, performance appraisal, team-based work, information-sharing programmes, training and development, career opportunity, and socializing activities [40–47].

This managerial turn led the scientific literature to focus on the cognitive, affective, and behavioral determinants of employees' perception of HRM practices. Accordingly, many positive individual and organizational outcomes were found to be significantly related to a high-commitment people management strategy. Yet, HRM practices were proved to influence very important aspect of the person/organization relationship such as, among others, job satisfaction, affective commitment, and retention intention [36,48,49], knowledge-sharing behavior [50], innovation activities [48], organizational citizenship behavior [51], employee creativity [52], job embeddedness [53], and work engagement and commitment [54].

A common conclusion drawn by most of these contributions is the evidence according to which the more the HRM system adopts practices and policies aimed to convey to employees a sense of

involvement, recognition, empowerment, competence development, fair rewards, and information sharing, the more this managerial choice could contribute to developing a person/organization relationship based on mutual trust and exchange thus influencing employees' behaviors and organizational performance.

Based on these speculations, the present study maintained that:

**Hypothesis 1a (H1a)**. *HRM involvement has a positive effect on organizational engagement*.

**Hypothesis 1b (H1b)**. *Extra-role behavior*.

*2.2. Sustainable HRM and Employee Change Attitude*

Organizational change is a constant condition of post-modern organizations. Accordingly, organizational change— defined as a number of alterations to the existing work routines and strategies that affect a whole organization [55]—has become a central focus in the strategic and change management literature [56–58].

During the COVID19 pandemic, organizations have been forced to ask their employees for a significant extra-effort toward change. The use of extraordinary measures, such as layoffs, the reduction of working hours, the introduction of smart and remote working contributed to reorganize work processes, job contents, formal and informal relationships within the workplace, heavily impacting upon employees' attitude and behavior, in terms of perceived role ambiguity, job insecurity, and social isolation among the workforce as showed by a priori research in the field [59,60].

Despite these evidences, in order to survive in such a fast-changing scenario, organizations need people to accept and manage change. Research estimated that in the 50% of cases, organizational change failed to deliver expected results and/or to meet intended objectives mostly because it had difficulties in committing employees to change [61–63]. A number of motivations could be found to justify such a high failure rate, however most management studies agreed to conclude that employees always play a major role within this process, holding the main responsibility for failure or success [64–66]. In line with this evidence, prior empirical studies confirmed that employees' resistances and reactions to change as well as their individual post change-related attitudes and behaviors could be considered crucial factors within the change-management process, influencing its course either positively or negatively [67–69].

As such, employees' commitment to change—defined as "a force (mind-set) that binds an individual to a course of action deemed necessary for the successful implementation of a change initiative" ([70] p. 475)—received growing attention as an important antecedent of change-related organizational outcomes [70,71]. Generally, employees are quite reluctant to organizational change, because it could be connected to an increase in workloads, to the assignment of new tasks on the top of the existing ones, to the need to adjust to new work relationships, and very often, to the introduction of new strategic goals [72,73]. However, when positively introduced by the organization change could be perceived as an opportunity for development, thus creating positive attitudes and consequently a positive effect on organizational performance [74,75].

Therefore, the main question is how to promote a positive attitude toward change in employees. Literature on commitment to change suggested that this attitude is mainly linked to the support that managers provide to change management and implementation [76]. In other words, the extent to which employees believe that organizational change could be a benefit, an opportunity, and a chance for development depends upon the perception of the consequences and implications that they think change would have for their personal experience within the organization [55]. Accordingly, to involve employees in the definition of a change management plan could be useful to create positive feelings and attitudes toward this behavioral effort, implying a cognitive and affective reframing of most certainties within the organizational context [77].

Abundant empirical evidences could be found in support of such assumption. Jorgensen and colleagues [78] found that a broader inclusion of employees and a strong culture of empowerment

and distributed leadership could be crucial factors responsible of a successful change management process. In a similar vein, Dent and Powley [79] suggested that employees tended to be more positive toward change if actively involved in its implementation. Chawla and Kelloway [80] contended that employees' participation to change could positively impact on the development of organizational trust and Weller and Bernadine [81] further supported the idea that successful change processes are associated with the degree and quality of employees' participation to these actions.

Likewise, another important variable explored by the literature on organizational change is related to employees' coping behavior namely any "conscious psychological and physical efforts to improve one's resourcefulness in dealing with stressful events...or to reduce external demands" [82] p. 45. Several scholars maintained the crucial role played by the coping strategies adopted to adjust to change and to manage its most distressful consequences, e.g., uncertainty, anger, stress, and conflict at work and at home [83].

Accordingly, further scientific contributions focused on the organizational variables that could enhance coping and commitment to change. Among others high-quality relationships between employees and managers [84], effective leadership practices [55,85,86], adequate technology/infrastructure to support change [86], and satisfaction with HR practices [87] were proved to be significantly associated with positive employees' attitudes and behaviors. Moreover, also some specific features related to the change management process were showed to play an important role. Yet, participation to the change process and perception of its fairness [88,89], adequate information and communication about change [86], and procedural and interactional justice of the change process [90] were found positively related to commitment and coping with change.

Consistent with these premises, the present study aimed to consider the positive relationship between HRM employee involvement and coping with organizational change, assuming that, during periods of change like the one generated by the COVID19 pandemic, those organizations that would adopt HR practices based on employee involvement and participation will be more likely to generate positive attitudes and behaviors toward change and its demands.

**Hypothesis 2 (H2)**. *HRM involvement has a positive effect on coping with organizational change*.

In light with these speculations, it is evident that the relationship between employees' perception of HRM policies and practices and their coping behaviors is very important for a positive management of change, because it is through employees' attitudes and behaviors that the organization might realize its proposed transformations, holding them successfully over time [76,91,92].

This perspective found several empirical evidences in the scientific literature. For instance, Cunningham [93] found that confidence in the ability to cope with organizational change was positively related with readiness to change, resulting in a stronger perception of the personal contribution given to change as well as in a higher degree of active participation to the whole process. Additionally, Judge, Thoresen and Pucik [94] found that the coping behavior was associated with several career outcomes, including salary, organizational commitment, satisfaction, and job performance. These studies suggested that employees who successfully cope with change initiatives are more likely to contribute to the process and to realize their own desired career outcomes. Furthermore, the context of change as well as employees' perception of HRM policies and practices were showed to be significant factors influencing individual reactions toward change [95,96] as coping with change, and the interaction between them could further positively impact on some specific employees' attitudes and behaviors, such as adaptive behaviors coping with unforeseen circumstances [97], citizenship and extra-role behavior [98], co-worker assistance [99], and change-supportive behavior [100] defined as "actions employees engage in to actively participate in, facilitate, and contribute to a planned change initiated by the organization" ([101], p. 1665). Following to these evidences, maintaining that employees who feel supported by the organization during change processes tend to hold a positive relationship with it and to perform coping behaviors with change itself, we offered our final hypothesis:

**Hypothesis 3a (H3a).** *HRM involvement shows an indirect positive relationship with organizational engagement and extra-role behavior, mediated by coping with organizational change.*

**Hypothesis 3b (H3b).** *When HRM involvement perception improves, organizational engagement and extra-role behavior increase, through an increase in coping with organizational change.*

## 3. Materials and Methods

### 3.1. Procedure and Participants

Participants to the study were a convenience sample of 549 Italian workers; researchers contacted participants asking them to fill in an online self-report questionnaire. The questionnaire's cover sheet explained the anonymity (of both participants and their organizations), confidentiality, and voluntariness of participation in the research. All participants provided their informed consent. The research observed the Helsinki Declaration [102] and the General Data Protection Regulation. The sample is composed of the dependent employees that complete the questionnaire during the lockdown imposed by the Italian government for a COVID19 pandemic emergency. Among the 547 participants, 62.1% were female and 37.7% were males. The mean age was 38.9 years (SD = 11.1). Regarding education, 32.9% had a high school diploma; 40.1% had a bachelor's or master's degree; 13.4% had a qualification higher than master's degree; 6% had a middle school diploma. Most of participants were married or cohabited (76.3%); 57.2% did not have children. The job contract was permanent for the 71.6% and fixed term for the 21.4%. Many of the participants were white collar workers, (50.3%), 13.8% were workers and 12.8% were manager and executive manager. They mostly belong to private (49.5%) and public (40.3) organizations and were from different occupational sectors: 28.6% tertiary sector; 16.6% education; professional services 14.1%; 10.6% secondary sector; 7.1% health; 6% primary sector; 16% other sectors. During the COVID19 emergency, 63.9% were working from home.

### 3.2. Measures

The variables investigated by the study were all measured using five-point Likert scales, with a score of 1 indicating very weak agreement with the item statement, and a 5 very strong agreement.

Employees' Involvement Perception in Human Resources Management Practices. This variable was measured by using 5 items of the subscale "Involvement" taken from the Human Resources Management Policies and Practices Scale (HRMPPS) elaborated by Demo, Neiva, Nunes, and Rozzett [103]. An example of the items used is "The organization I work for is concerned with my well-being". Cronbach's alpha was 0.91.

Employees' coping with change. This variable was assessed by adopting three items taken from the coping with organizational change scale [94]. An example of the item is "I see the rapid changes that are occurring in this company as opening up new career opportunities for me." Cronbach's alpha was 0.70.

Organizational engagement. Three items from the scale longest version of the measure elaborated by Saks [104] were considered to measure this variable (e.g., "I am highly engaged in this organization"). Cronbach's alpha was 0.85.

Extra-role behavior. Two items were considered from the extra-role behavior scale elaborated by Podsakoff, MacKenzie, Moorman, and Fetter [105]. An example of the items is "I help others at this organization with their responsibilities." Cronbach's alpha was 0.76.

### 3.3. Data Analysis

The statistics software SPSS 26 (IBM, Armonk, NY, USA) was used to test the descriptive data analysis (mean and standard deviation), Pearson correlations, and Cronbach's alpha coefficients while Mplus 7 (Muthén & Muthén, Los Angeles, CA, USA) supported the test of study hypotheses through

a Structural Equation Model (SEM). Maximum Likelihood (ML) was the estimation method and we controlled for job insecurity and remote working (1 = remote working at least 2 days per week; 0 = not remote working). The bootstrapping procedure tested the significance of the indirect effects, with the extraction of 2000 new samples from the original one [106]. The following goodness-of-fit criteria were examined to assess the model [107]: the $\chi^2$ goodness-of-fit statistic; the Root Mean Square Error of Approximation (RMSEA); the Comparative Fit Index (CFI) and the Tucker–Lewis Index (TLI); the Standardized Root Mean Square Residual (SRMR).

## 4. Results

Table 1 shows the means, standard deviations, and correlations between the study variables. Both organizational engagement and extra-role behavior positively correlated with HRM involvement and coping with change. Job insecurity negatively correlated with all the other variables while remote working showed a positive correlation with organizational engagement, extra-role behavior, and HRM involvement.

**Table 1.** Means, standard deviations, Cronbach's Alphas (on the diagonal), and correlations among study variables (N = 549).

| Variables | 1 | 2 | 3 | 4 | 5 | 6 |
|---|---|---|---|---|---|---|
| **1. Organizational engagement** | 0.85 | | | | | |
| **2. Extra-role behavior** | 0.54 ** | 0.76 | | | | |
| **3. Coping with change** | 0.52 ** | 0.40 ** | 0.69 | | | |
| **4. HRM involvement** | 0.60 ** | 0.32 ** | 0.36 ** | 0.92 | | |
| **5. Job insecurity** | −0.19 ** | −0.12 ** | −0.25 ** | −0.13 ** | - | |
| **6. Remote working (1 = yes)** | 0.09 * | 0.04 | 0.17 ** | 0.15 ** | −0.15 ** | - |
| *M* | 3.86 | 4.16 | 3.10 | 3.52 | 2.08 | - |
| *SD* | 0.92 | 0.87 | 0.91 | 0.93 | 1.24 | - |

*Note.* ** $p < 0.01$; * $p < 0.05$.

Figure 1 shows results of the SEM, which adequately fitted to the data: $\chi^2$ (81) = 261.30, $p < 0.001$, CFI = 0.95, TLI = 0.94, RMSEA = 0.06 (90% CI: 0.05, 0.07), SRMR = 0.05. Table 2 presents the results of alternative models which confirmed that the hypothesized model with the mediation of coping with change is the best solution. In the final model, HRM involvement showed a positive relationship with organizational engagement ($\beta$ = 0.51; $p < 0.001$), extra-role behavior ($\beta$ = 0.21; $p < 0.001$), and coping with change ($\beta$ = 0.39; $p < 0.001$). In turn, coping with change was positively associated with both organizational engagement ($\beta$ = 0.41; $p < 0.001$) and extra-role behavior ($\beta$ = 0.42; $p < 0.001$). Job insecurity was negatively associated with both HRM involvement ($\beta$ = −0.12; $p$ = 0.006) and coping with change ($\beta$ = −0.23; $p < 0.001$) while remote working showed a positive relationship with HRM involvement ($\beta$ = 0.14; $p$ = 0.002) and a not significative relationship with coping with change.

**Table 2.** Results of alternative Structural Equation Models (SEMs).

| Models | $\chi^2$ | df | p | CFI | TLI | RMSEA | SRMR | Comparison | $\Delta\chi^2$ | p |
|---|---|---|---|---|---|---|---|---|---|---|
| **M$_1$.** | 261.30 | 81 | <0.001 | 0.95 | 0.94 | 0.06 (0.05, 0.07) | 0.05 | | | |
| **M$_2$.** | 318.68 | 82 | <0.001 | 0.94 | 0.92 | 0.07 (0.06, 0.08) | 0.10 | M$_2$–M$_1$ | 57.38 | <0.001 |

*Note:* M$_1$ is the hypothesized model with coping with change as mediator. M$_2$ is the direct effects model without mediation of coping with change.

The model explained about 62% of the variation in organizational engagement, 29% in extra-role behavior, 26% in coping with change, and only 4% in HRM involvement. Table 3 shows the statistically significant indirect effects, results of the bootstrapping procedure. In particular, a positive indirect effect was confirmed between HRM involvement and both organizational engagement and extra-role behavior through the mediation of coping with change.

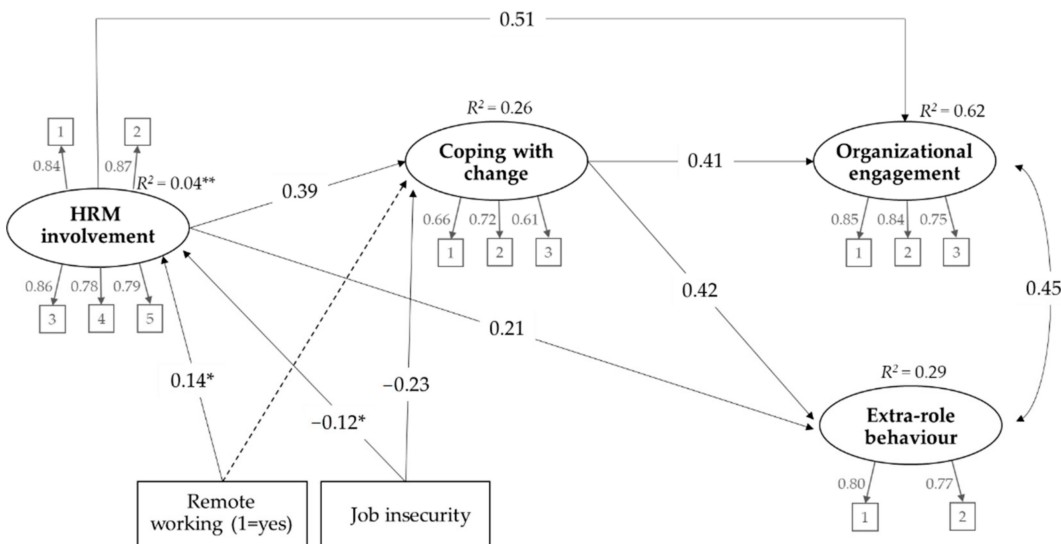

**Figure 1.** The hypothesized model controlling for job insecurity and remote working (Maximum Likelihood (ML) estimation; standardized path coefficients; N = 549). Discontinuous line indicates a non-significant relationship. $p < 0.001$; * $p < 0.01$; ** $p < 0.05$.

**Table 3.** Indirect effects using the bootstrapping procedure (2000 replications).

| Indirect Effects | Est. | S.E. | *p* | CI 95% |
|---|---|---|---|---|
| HRM inv. → Cop. → Org. Eng. | 0.16 | 0.04 | <0.001 | (0.10, 0.25) |
| HRM inv. → Cop. → Extra-role | 0.16 | 0.05 | 0.001 | (0.09, 0.27) |
| Job. Ins. → Cop. → Org. Eng. | −0.10 | 0.02 | <0.001 | (−0.15, −0.05) |
| Job. Ins. → Cop. → Extra-role | −0.10 | 0.03 | <0.001 | (−0.15, −0.05) |
| Job. Ins. → HRM inv. → Org. Eng. | −0.06 | 0.02 | 0.011 | (−0.11, −0.02) |
| Job. Ins. → HRM inv. → Extra-role | −0.03 | 0.01 | 0.043 | (−0.05, −0.01) |
| Job. Ins. → HRM inv. → Cop. → Org. Eng. | −0.02 | 0.01 | 0.026 | (−0.04, −0.01) |
| Job. Ins. → HRM inv. → Cop. → Extra-role | −0.02 | 0.01 | 0.046 | (−0.04, −0.01) |
| Rem. Work → HRM inv. → Org. Eng. | 0.07 | 0.02 | 0.002 | (0.03, 0.12) |
| Rem. Work → HRM inv. → Extra-role | 0.03 | 0.01 | 0.022 | (0.01, 0.06) |
| Rem. Work → HRM inv. → Cop. → Org. Eng. | 0.02 | 0.01 | 0.024 | (0.01, 0.05) |
| Rem. Work → HRM inv. → Cop. → Extra-role | 0.02 | 0.01 | 0.038 | (0.01, 0.05) |

*Note:* All parameter estimates are presented as standardized coefficients. Estimates (Est.). Standard Error (SE). Confidence Interval (CI). HRM involvement (HRM inv.). Coping with change (Cop.). Organizational engagement (Org. Eng.). Extra-role behavior (Extra-role). Job insecurity (Job. Ins.). Remote working (Rem. Work).

## 5. Discussion

The aim of the study was to investigate the role played by coping with organizational change in the relationship between HRM involvement perception and employees' positive organizational behaviors, namely organizational engagement and extra-role behavior, during the COVID19 pandemic emergency. Results gave very impressive suggestions that could be useful to develop future research and to draw practical implications in the field of human resources management, with special reference to crisis management.

The examination of the research model fully met the study hypotheses. Hypothesis 1 was confirmed: HRM involvement perception was proved to be positively related to organizational engagement (H1a) and to extra role behavior (H1b). These results pointed out that the presence of high levels of employees' HRM involvement perception could improve organizational engagement,

increasing employees' extra role behavior. Specifically, if employees experience support coming from a high-involvement HRM system (e.g., working in teams, participating in upward feedback programs, and receiving information about organizational results) then they would probably engage more in positive behaviors [13,24–26]. This evidence is further confirmed by studies adopting a cross-level analysis of the relationships between high-involvement HR systems and employee attitudes and behaviors, such as job satisfaction, employee commitment, and absenteeism, comparing self-report descriptions of individual perceptions about HRM and more objective measures of the practices used by work systems [108–111].

Furthermore, HRM involvement perception was positively associated with coping with organizational change, confirming hypothesis 2. Therefore, HRM practices in the form of rewards, training, performance management, task autonomy, job enrichment, and involvement would promote positive attitude toward change of employees by increasing their enthusiasm, optimism, compliance, participation, and collaboration for the change [112]. Results showed that employees' attitudes and commitment to change, the strategies they adopt to promote, to disseminate, and to support change largely depends on the way by which the organization communicates, involves, conveys information, accompanies, and supports employees in the process of change. At the same time, employees' efforts toward change have been linked to organizational citizenship behaviors, such as extra-role behavior, that can be considered as behavioral indicators of support for organizational change [113,114]. Organizational citizenship behaviors are discretionary actions on the part of employees that benefit the organization and they represent a behavioral index of employee alignment with the change initiative [113,114]. The sense that positive changes are occurring is likely to create greater energy or enthusiasm around the change initiative. In line with recent studies, results suggest that when employees perceive positive changes occurring in their organization, they tend to feel more committed to change and consequently engage in efforts and behaviors that might concretely sustain change. Specifically, a positive perception of change might increase energy feelings around change, therefore inspiring employees to be more committed and engaged [68,115–117].

Finally, HRM involvement perception of employees showed a positive relationship with organizational engagement and extra-role behavior, through the partial mediation of coping with change (H3a and H3b). This evidence confirmed that, although HRM involvement inspire positive organizational behavior, namely organizational engagement and extra-role behavior, employees' positive attitude toward organizational change, and the related coping strategy adopted, contribute significantly to increasing this impact.

The study also provided relevant results related to the remote working condition and to job insecurity, both considered as a control variable in the model. First, findings suggested a negative relationship between job insecurity and HRM involvement and coping with change. At the company level, company performance, management changes, as well as formal and informal announcements of impending changes serve as more proximal warning signs that one's job might be at risk and are accordingly associated with increased job insecurity [118–120]. Therefore, especially during a period of great uncertainty, like the one featured by the current pandemic, it becomes essential to adopt human resource management systems and practices that could contribute to reassure employees about their job security, supporting them in accepting and positively committing to change. Likewise, job insecurity is negatively associated with coping with organizational change. This result was in line with previous research assuming that job insecurity perceptions might lead to resistance to change, whereas job security perceptions, on the other hand, could foster openness to change attitudes and behaviors [80]. This finding could be very interesting for future research investigating the critical aspect of this important individual factor that might impact the change management processes.

Finally, the research model underlined a positive relationship between remote working and HRM involvement. This quite unexpected result could be probably explained by the many efforts accomplished by companies during the period when the study was conducted to empower HRM systems, trying to intensify for instance communication and information practices to keep control over

diffused feelings of fear and anxiety, over the sense of isolation and insecurity experienced by workers while switching to remote work. This intensification of a people-based approach to HRM might have conveyed a higher sense of involvement that finally positively impacted the positive behaviors.

Therefore, the challenge of the engagement of remote workers is one of the main tasks of HRM systems during the pandemic. Indeed, remote working in this case is not an option, it is forced by the contextual circumstances (in response to COVID-19) and therefore it demands an extra-effort in terms of employees' motivation and commitment to change, both positive behaviors that HR systems could fruitfully manage as the present study showed.

## 6. Limitations, Practical Implications, and Conclusions

Despite the contribution given by the study in the field, some limitations could be also highlighted. A first limitation was related to the cross-sectional nature of the study, that did not allow neither to test eventual causal relationship among variables nor to investigate them across time. Certainly, also the exceptional historical and social moment when data collection was carried out (during the COVID19 pandemic) might have influenced employees' responses and/or the way HR policies and practices might have been communicated and managed influencing the person/organization relationship.

Another limitation could be found in the limited and heterogenous sample involved that cannot allow the generalization of results. The convenience sampling and the collection procedure that used an online questionnaire were aimed to involve employees belonging to different kind of professional categories (e.g., collar workers, workers, executive manager), working in different occupational sectors, in private and public contexts, and having different kind of employment contracts with the organization (e.g., permanent and fix-term). These variables could have played an important role with special reference to the HRM policies and practices adopted within different organizational contexts and with reference to different professional target groups. Therefore, future research should consider replicating the study by adopting longitudinal and/or diary methodologies and by focusing attention on a specific occupational sector and/or professional category.

Finally, another possible limitation of the study was related to the risk of common-method bias [121] and to the self-report measures used to assess the variables investigated. The first risk is generally associated with the use of common methods to measure multiple constructs (e.g., multiple-item scales presented within the same survey as in the present study) leading to spurious effects due to the measurement instruments rather than to the constructs being measured. This limitation could be addressed by future research adopting different scale endpoints and formats for the measures used. On the other hand, in order to overcome biases implied in the use of self-report measures, more objective checklists and observation tools could be integrated to collect and analyze the typology of HRM practices confronting this information with employees' perceptions about them and finally with their consequent positive behaviors.

Besides the limitations highlighted above, results from the study pointed out a few possible implications in terms of managerial practices.

First, the topic chosen, which is inscribed within the wider crisis and change management literature, contributed to further discuss the evidences about the impact of a people-based approach to HRM on employees' attitudes and performance both in terms of subjective coping strategies (coping with change) and organizational behaviors (extra-role behaviors and organizational engagement) especially during this challenging time of pandemic.

Actually the outbreak of COVID19 forced many organizations to make decisions rapidly and efficiently especially about human resource management, deciding for instance who should stay at work and who should go home (in smart working or in payroll subsidies); what kind of processes and tasks could be moved into a digital space; and which were the organizational priorities and how these priorities could be best communicated to employees [122]. Undoubtedly, this emergency had a significant economic impact on the labor market as well as an important psychological effect on

individuals and organizations. Venkatesh [123] suggested that there are at least two main categories of effects that the pandemic had on workers: effects on the job and effects on life.

The category of the effects on the job includes: Job loss—many people lost their jobs in many countries [124]; job changes—many workers experienced a substantial change in contents and modalities of their job, with an unprecedent extent of remote work and with its consequent effects on employees anxiety, technostress, and overload [125]; job outcomes—decreasing performance levels, job satisfaction, organizational commitment, and increasing job stress, coping strategies, and support requests. Likewise, the category of the effects on life encompasses many other important aspects that workers have been called to manage. Evidently the pressure applied by COVID19 on individuals, communities and organizations posed urgent issues to address even within the life domain: home life changes, managing children, life-related outcomes, social life conditions, and the need of support are just some of the features that constantly interact with the changes occurred in the working life.

In line with this scenario, results suggested the need to re-focus on the centrality of the human resources management function within organizations, redefining policies and practices of people management in order to support employees in coping with this very difficult moment of uncertainty, further exacerbated by the fear for the health consequences of the COVID19 pandemic.

In this time of great change HRM should make use of its most strategic levers such as training and development programs in order to reinforce employees' identification, motivation, and engagement to the organizational project [126,127]. To this purpose, HRM could rely upon some of the most strategic factors of the organization such as leadership, talent management, skill development, and communication [122]. All efforts should be addressed to re-establish the psychological contract between individual and organization, to create a positive climate of mutual acknowledgement and trust that could encourage employees' engagement toward change, their performance and consequently organizational success [128,129]. Accordingly, the challenges posed by the current global uncertainty amplified by the pandemic seemed to further confirm the crucial role played by HRM in organizations, being a partner in the strategic management of the people and processes, that concretely substantiate organizational effectiveness [130].

In this vein, results coming from the present study attempted to give a contribution in this direction, confirming the value chain that leads to sustainable development: to invest in HRM policies and practices is to invest in the human capital, namely in the intangible asset of individuals' resources (knowledge, talents, skills, experience), that if effectively managed and developed could turn into positive attitude and behaviors and therefore into an actual competitive advantage for organizations.

**Author Contributions:** Conceptualization, A.M., M.L.G., M.M., E.I., F.S., V.R., M.Z., and C.G.C.; methodology, M.M., M.L.G., A.M., E.I., F.S., V.R., M.Z., and C.G.C.; validation, A.M., M.L.G., M.M., E.I., F.S., V.R., M.Z., and C.G.C.; formal analysis, M.M.; investigation, A.M., M.L.G., M.M., E.I., F.S., V.R., M.Z., and C.G.C.; data curation, M.M.; writing—original draft preparation, A.M., M.L.G.; writing—review and editing, A.M., M.L.G., M.M., E.I., F.S., V.R., M.Z., and C.G.C.; visualization, A.M., and M.L.G.; supervision, E.I., A.M., V.R., and C.G.C.; project administration, E.I. and C.G.C. All authors have read and agreed to the published version of the manuscript.

**Funding:** This research received no external funding.

**Conflicts of Interest:** The authors declare no conflict of interest.

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
