# Peer review of "“Everything Will Be Fine”: A Study on the Relationship between Employees’ Perception of Sustainable HRM Practices and Positive Organizational Behavior during COVID19"

_sustainability, doi:10.3390/su122310216_

Round 1
Reviewer 1 Report
The topic is very interesting. The authors did a good attempt to project the need to re-focus on the centrality of the HR management function within organizations and to redefine policies and practices to support employees in moments of uncertainty. However, in my view the discussion needs improvement. You mentioned positive changes, positive relationship. It would be good if you mention strategies with the support of literature in the discussion.
Author Response
We are very grateful to your valuable comment about the need to better specify the discussion and conclusion because it gave us the opportunity to further go into details about the crucial role of HRM in change management processes. The integrations are written in red in the manuscript uploaded.

Reviewer 2 Report
This article deals with very important situation in current time period. I recommend the article for publication in submitted form.I positively evaluated all parts of the scientific article. I hope that the results will have not only theoretical, but also practical contribution for readers in all spheres.
Author Response
We would like to thank you very much for the positive feedback to our work. We have tried to integrate the conclusions, in order to follow your comments about the importance of highlighting concrete implications of the study as well as the suggestions coming from reviewer 1.
Reviewer 3 Report
-
Author Response
We did not found any comment added to the evaluation. We hope this is a positive feedback about the appreciation of our work.